# TCO, a Putative Transcriptional Regulator in *Arabidopsis*, Is a Target of the Protein Kinase CK2

**DOI:** 10.3390/ijms20010099

**Published:** 2018-12-28

**Authors:** Laina M. Weinman, Katherine L. D. Running, Nicholas S. Carey, Erica J. Stevenson, Danielle L. Swaney, Brenda Y. Chow, Nevan J. Krogan, Naden T. Krogan

**Affiliations:** 1Department of Biology, American University, 4400 Massachusetts Avenue NW, Washington, DC 20016, USA; laina.weinman@alumni.american.edu (L.M.W.); katherine.running@ndsu.edu (K.L.D.R.); nc7356a@student.american.edu (N.S.C.); bchow@american.edu (B.Y.C.); 2Gladstone Institute of Data Science and Biotechnology, J. David Gladstone Institutes, San Francisco, CA 94158, USA; erica.stevenson@gladstone.ucsf.edu (E.J.S.); danielle.swaney@ucsf.edu (D.L.S.); nevan.krogan@ucsf.edu (N.J.K.); 3Quantitative Biosciences Institute (QBI), University of California, San Francisco, CA 94158, USA

**Keywords:** *Arabidopsis thaliana*, chromatin regulation, CK2, development, phosphorylation, protein kinase, transcriptional regulation

## Abstract

As multicellular organisms grow, spatial and temporal patterns of gene expression are strictly regulated to ensure that developmental programs are invoked at appropriate stages. In this work, we describe a putative transcriptional regulator in *Arabidopsis*, TACO LEAF (TCO), whose overexpression results in the ectopic activation of reproductive genes during vegetative growth. Isolated as an activation-tagged allele, *tco-1D* displays gene misexpression and phenotypic abnormalities, such as curled leaves and early flowering, characteristic of chromatin regulatory mutants. A role for TCO in this mode of transcriptional regulation is further supported by the subnuclear accumulation patterns of TCO protein and genetic interactions between *tco-1D* and chromatin modifier mutants. The endogenous expression pattern of *TCO* and gene misregulation in *tco* loss-of-function mutants indicate that this factor is involved in seed development. We also demonstrate that specific serine residues of TCO protein are targeted by the ubiquitous kinase CK2. Collectively, these results identify TCO as a novel regulator of gene expression whose activity is likely influenced by phosphorylation, as is the case with many chromatin regulators.

## 1. Introduction

Growth and development in multicellular eukaryotes involves the integration of diverse cell types whose specification is under strict spatial and temporal control. This fate specification is the product of precise gene expression patterns established through a balance between transcriptional activation and repression. On a broad scale, this balance ensures that developmental programs are invoked at appropriate stages in an organism’s life cycle. While eukaryotes employ many mechanisms to mediate this control, chromatin modification is especially well-suited to establish and sustain a particular transcriptional state. For example, Polycomb group (PcG) and Trithorax group (trxG) proteins are chromatin modifiers that play critical roles by stably repressing or activating transcription, respectively. Two important PcG multi-protein complexes include PRC2, which trimethylates histone H3K27, and PRC1, which ubiquitylates histone H2A to compact chromatin and silence gene expression [1]. Conversely, trxG proteins antagonize PcG repression through mechanisms that include trimethylation of histone H3K4 and ATP-dependent chromatin remodeling [1].

In higher plants, reproductive flowers arise late in development, with floral-specific genes being repressed throughout vegetative growth. In the model angiosperm *Arabidopsis thaliana*, chromatin modifiers are key to this repression, as their mutation results in ectopic gene expression and numerous developmental defects. For example, mutation of the histone methyltransferase *CURLY LEAF* (*CLF*), a PRC2 component, causes derepression of floral organ genes and results in upward-curled leaves, small rosettes, early flowering, short inflorescence stems, and homeotic conversions of floral organs [2]. Antagonizing *CLF* function are trxG factors such as *ARABIDOPSIS HOMOLOG OF TRITHORAX1* (*ATX1*) and *ULTRAPETALA1* (*ULT1*), mutations of which suppress *clf* defects [3,4]. Consistent with this antagonistic relationship, *Arabidopsis* plants overexpressing *ULT1* display similar phenotypic abnormalities as *clf* loss-of-function mutants [4]. Apart from floral organ gene repression, different *Arabidopsis* PRC2 complexes control diverse growth programs, ranging from endosperm and seed development to vernalization-dependent floral transition [5].

Since organismal growth programs can be strongly influenced by dynamic internal and external cues, they are often plastic in nature [6,7]. It follows that the chromatin regulation underlying these processes is similarly flexible. Post-translational protein modification is an effective strategy for conferring such plasticity, and chromatin regulators are commonly subject to these modifications, including phosphorylation [8]. Phosphorylation can influence many protein properties such as subcellular localization, enzymatic activity, protein–protein interactions, and chromatin association [9]. For example, phosphorylation of the PRC2 methyltransferase Enhancer of Zeste homolog 2 (EZH2) prevents histone binding and compromises its catalytic activity, resulting in gene derepression [10]. Human orthologs of the trxG SNF/SWI complex are also phosphorylated during mitosis, resulting in their exclusion from chromatin [11].

Phosphorylated chromatin modifiers often harbor kinase recognition sequences that are acidic in nature, implicating casein kinase 2 (CK2), a ubiquitous Ser/Thr protein kinase, as a prevalent regulator [8]. Indeed, the *Drosophila* PRC2 component Extra sex combs (Esc) and its mammalian ortholog Embryonic Ectoderm Development (EED) are phosphorylated by CK2, which promotes protein homodimerization and influences overall complex formation [12,13]. Similarly, CK2-mediated phosphorylation of the PRC1 subunit Cbx2 in mouse changes its binding affinity for modified histone H3 [14].

In the present work, we describe the isolation of an activation-tagged mutant in *Arabidopsis* we named *taco leaf-1D* (*tco-1D*) due to its strongly upward-curled leaves. Numerous properties of *tco-1D* are consistent with disrupted chromatin regulation, including its developmental abnormalities, misregulated genes, and genetic interactions with mutants of PcG and trxG homologs. Additionally, the nuclear localization of TCO often displays a “speckled” distribution reminiscent of numerous eukaryotic chromatin regulators. The expression pattern of *TCO* and defects associated with *tco* loss-of-function mutants implicate this factor in the regulation of seed development. Finally, TCO is bound and phosphorylated by CK2, suggesting that a posttranslational regulatory mechanism broadly used to control the activity of chromatin modifiers may also influence TCO function.

## 2. Results

### 2.1. Identification and Characterization of the Leaf-Curling Mutant Taco Leaf-1D

To identify novel genes involved in lateral organ patterning, we performed an activation-tagging mutant screen in *Arabidopsis*. We isolated one dominant mutant that exhibited strong upward leaf curling (Figure 1A–D) that we named *taco leaf-1D* (*tco-1D*). Apart from this leaf-curling defect, which affected both rosette and cauline leaves (Figure 1B,D,F), *tco-1D* flowered earlier than wild type based on the number of rosette leaves initiated prior to the floral transition (Figure 1G). The leaves and inflorescence stems of *tco-1D* were also shorter than those of the wild type (Figure 1H,I), giving the mutant a highly compact stature (Figure 1E) with a reduced number of siliques (Figure 1J).

Using a thermal asymmetric interlaced (TAIL) polymerase chain reaction (PCR)-based approach [15], we determined that the activation-tagging T-DNA in *tco-1D* resides between genes *At4g23110* and *At4g23120* in the *Arabidopsis* genome, with the tandemly repeated 2x35S CaMV enhancers of the T-DNA oriented towards *At4g23110* (Figure 2A). When semi-quantitative reverse transcription-PCR (RT-PCR) was used to test *At4g23120* transcript levels in *tco-1D* and wild-type seedling tissue, expression could not be detected in either background, despite efficient PCR amplification using the same primers and genomic DNA as template (Figure 2B). Expression of *At4g23110* was also undetectable in wild-type seedling tissue; however, transcription of this gene was clearly upregulated in *tco-1D* (Figure 2B). These results were verified by quantitative RT-PCR, which showed that *At4g23110* was ectopically expressed in *tco-1D* vegetative tissues in a reliable and reproducible fashion, while *At4g23120* was not (Appendix A). Collectively, these observations implicate *At4g23110* as the activation-tagged locus in *tco-1D*.

To verify this prediction, we used the constitutive 2x35S CaMV promoter to drive expression of an At4g23110-GFP translational fusion (2x35Sp::TCO-GFP) and found that this transgene phenocopied *tco-1D* (Figure 2C,D,H). Therefore, we conclude that the identity of the *TCO* gene is *At4g23110*, which encodes a small (148 amino acid) protein sharing some homology with animal insulin-like growth factor binding proteins (as annotated by TAIR (www.arabidopsis.org)) [16]. *BLAST* alignment searches (www.ncbi.nlm.nih.gov/blast) [17] of other plant genomes identified proteins similar to TCO/At4g23110 in other *Brassicaceae* species, including *Arabidopsis lyrata*, *Capsella rubella*, *Camelina sativa*, *Eutrema salsugineum*, *Raphanus satvius* and *Brassica nap*us (Appendix A).

To gain insight into the function of TCO, we transiently expressed 2x35Sp::TCO-GFP in tobacco leaves to determine its subcellular localization. TCO protein localized to the nucleus, including in a subcellular compartment that resembled the nucleolus (Figure 2E–G). Bright TCO-GFP foci or “speckles” were also apparent in some nuclei (Figure 2G). Expression of 2x35Sp::TCO-GFP in stable *Arabidopsis* transformants also displayed punctate subcellular accumulation patterns consistent with nuclear localization (Appendix A). These results indicate that TCO may function in transcriptional regulation and that the pleiotropic defects of *tco-1D* may result from widespread gene misexpression.

### 2.2. Reproductive Genes Are Ectopically Expressed in tco-1D Vegetative Tissues

Short, upward-curled leaves and early flowering in *Arabidopsis* can result from ectopic expression of floral genes in vegetative tissues [2]. To test whether floral gene misregulation contributes to *tco-1D* developmental defects, we first crossed *tco-1D* to *agamous* (*ag*), a C-class floral organ identity gene as defined by the ABC model of floral patterning [18,19]. The rosette leaves of *tco-1D ag-1* double mutants were less curled and significantly longer than those of *tco-1D* alone (Figure 3A–E), suggesting that ectopic *AG* expression contributes to *tco-1D* developmental abnormalities. Consistent with this, quantitative RT-PCR confirmed that *AG* is upregulated approximately 6-fold in *tco-1D* seedling tissue compared to that of wild type (Figure 3F; Appendix A).

The incomplete suppression of *tco-1D* by *ag* indicates that other genes are likely misexpressed in this background as well. To test this hypothesis, we assessed the expression levels of other floral patterning and/or flowering time genes (Figure 3F; Appendix A), many of which result in leaf curling and/or early flowering when misregulated [20,21,22,23,24,25,26,27,28,29]. Numerous MADS-box floral organ identity genes were upregulated 2-fold or greater in *tco-1D*, including the A-class gene *APETALA1* (*AP1*), the B-class genes *AP3* and *PISTILLATA* (*PI*), the D-class genes *ARABIDOPSIS B SISTER* (*ABS*), *SHATTERPROOF2* (*SHP2*) and *SEEDSTICK* (*STK*), and the E-class gene *SEPALLATA3* (*SEP3*) (Figure 3F; Appendix A). Conversely, the expression of the D-class gene *SHP1* was not appreciably altered, indicating that the effect of *tco-1D* on floral MADS-box genes displays some specificity. Further, the floral meristem identity gene *LEAFY* (*LFY*) and the flowering time gene *FLOWERING LOCUS T* (*FT*) did not show increased expression levels (Figure 3F; Appendix A). This latter observation suggests that the early flowering of *tco-1D* is not a product of *FT* upregulation, but may be due to the upregulation of floral organ identity genes such as *SEP3* [26]. Collectively, these results show that the *tco-1D* mutation leads to the ectopic expression of numerous floral organ identity genes, and that at least some of this gene misregulation contributes to *tco-1D* developmental defects. Along with its nuclear localization, these observations are consistent with TCO playing a role in the control of gene expression.

### 2.3. TCO Displays Characteristics of a Chromatin Regulator

TCO protein often accumulates in punctate subnuclear foci or “speckles” (Figure 2F,G), which is reminiscent of numerous chromatin-modifying proteins in both animals and plants [4,30,31,32,33]. This localization pattern suggests that TCO may influence gene expression through chromatin regulation. Indeed, the developmental defects and gene misregulation exhibited by *tco-1D* closely resembles those of PcG loss-of-function mutants (*clf*) and trxG overexpression lines (35Sp::ULT1) [2,4]. This includes upward-curled leaves, small rosettes, early flowering, short inflorescence stems that terminate growth prematurely, and ectopic expression of floral organ identity genes including *AG*, *AP3* and *SEP3* [2,4]. Moreover, ULT1 has been reported to exhibit a speckled subnuclear distribution pattern similar to that displayed by TCO [4].

If TCO influences transcription via chromatin regulation, the developmental defects of *tco-1D* may be enhanced by mutation of *CLF* and/or suppressed by mutation of *ULT1*, which is known to antagonize *CLF* activity in *Arabidopsis* [4]. As a first test of this hypothesis, we crossed *tco-1D* with *clf-2* to test for genetic interaction. Cotyledons of *tco-1D* mutants display increased angles of upward growth compared to wild-type cotyledons (Figure 4A,B). This hyponastic response was markedly enhanced in the *tco-1D clf-2* double mutant, which produced cotyledons that commonly bent over the top of the seedling to completely enclose newly initiated vegetative leaves (Figure 4C,D). Furthermore, *tco-1D clf-2* seedlings displayed tightly curled leaves that were significantly shorter than either single mutant (Figure 4E–H,K). Conversely, the phenotypic null allele *ult1-2* [34] partially suppressed *tco-1D* leaf defects, as the *tco-1D ult1-2* double mutant leaves were less tightly curled and significantly longer than *tco-1D* leaves (Figure 4F,I–K). Similar results were obtained from crosses between the semi-dominant allele *ult1-1* [34] and *tco-1D* (Appendix A). Taken together, these genetic interactions suggest that TCO plays a role in chromatin regulation, potentially by interacting with or influencing the activity of PcG and/or trxG homologs in *Arabidopsis*.

### 2.4. TCO Exhibits Seed-Specific Expression

To determine the endogenous *TCO* expression pattern, we generated a TCOp::TCO-GUS reporter transgene under the regulatory control of 2.4 kb and 1.6 kb of upstream and downstream *TCO* non-coding sequence, respectively. Consistent with our inability to detect expression of *TCO* in vegetative tissues (Figure 2B), previous transcriptomic analyses indicated that *At4g23110* expression was seed-specific [35]; therefore, we concentrated our reporter analyses on this stage. Multiple independent transgenic TCOp::TCO-GUS lines showed expression throughout the seed, including in the seedcoat, endosperm and developing embryo (Figure 5A,B), suggesting a role for *TCO* in seed and/or embryo development.

To test whether such stages were affected by loss of *TCO* function, we analyzed two T-DNA insertion mutants that interrupted the *TCO* coding region (Appendix A). Unlike wild type, in both insertion lines (designated *tco-1* and *tco-2*), full-length *TCO* transcripts were undetectable in silique tissue shortly after pollination, as assessed by RT-PCR using primers spanning the T-DNA insertion sites (Appendix A). Furthermore, because both T-DNA insertions are positioned in the 5′ half of the *TCO* coding sequence (Appendix A), it is unlikely that either allele produces functional TCO protein. Because multiple seed genes were ectopically expressed in *tco-1D*, we tested whether similar gene misregulation was apparent in *tco* silique tissue. At 2 days after pollination (DAP), the expression of C- and D-class MADS-box seed genes in the insertion lines was similar to wild type (Figure 5C). However, at 4 DAP, both insertion lines displayed an increase in the expression of the D-class gene *ABS*, and subtle increases in other D-class genes (Figure 5D).

Consistent with a role for *TCO* in seed development, *tco-1* also displayed seed defects at incomplete penetrance. Specifically, at 9 DAP, a proportion of seeds in *tco-1* siliques were white-to-light green in coloration (32.1%, *n* = 661) (Appendix A). Tissue dissections revealed that these light-colored seeds harbored embryos at the globular stage, while their dark-colored siblings contained bent-cotyledon stage embryos, as did wild-type seeds of the same age (Appendix A). In comparison, *tco-2* exhibited seed defects at a much lower frequency (1.9%, *n* = 647) which was comparable to the wild type (1.2%, *n* = 1705). Although both *tco* insertions affected *TCO* expression (Appendix A), it is possible that seed defects are only prevalent in *tco-1* due to the more upstream position of its T-DNA (Appendix A). Finally, seed defects were not apparent in *tco-1D* or 2x35Sp::TCO-GFP plants, apart from an overall reduction in seed production due to their small stature (Figure 1E,I,J; Figure 2C,D). Thus, seed/embryo development may be more sensitive to reductions in *TCO* function compared to increased *TCO* levels.

### 2.5. TCO Interacts with and Is Phosphorylated by the Protein Kinase CK2

Because transcriptional regulators often rely on protein–protein interactions for their activity, we sought to better understand TCO function by screening for physical interactors. To this end, we performed a yeast two-hybrid (Y2H) screen using TCO as bait and an *Arabidopsis* floral cDNA library as prey. Of the 1.7 × 10^6^ yeast transformants isolated, 22 interactors were identified, all of which were clones of the catalytic subunit (α) of CK2. The CK2 holoenzyme is a tetramer of two α subunits and two regulatory β subunits, with each type of subunit encoded by four different genes in *Arabidopsis* [36]. All CK2 subunits are broadly expressed throughout *Arabidopsis* development (including in roots, rosette leaves, reproductive stems, and flowers) but show some variability in subcellular localization patterns [36]. For instance, α subunits encoded by *AtCKA1* (*At5g67380*), *AtCKA2* (*At3g50000*) and *AtCKA3* (*At2g23080*) localize to the nucleus (and are enriched in the nucleolus), whereas the protein product of *AtCKA4* (*At2g23070*) localizes to the chloroplast [36]. The 22 identified TCO interactors consisted of 4 AtCKA1 and 18 AtCKA2 clones. Notably, the reported subcellular localization of these α subunits matches that of TCO, which is also localized to the nucleus (Figure 2F,G). The protein products of the isolated *AtCKA1* and *AtCKA2* cDNA clones interacted strongly with TCO when re-tested in directed Y2H assays, even though each clone had a short truncation at the 5′ terminus of its coding sequence (Figure 6A). We also generated full-length *AtCKA1* and *AtCKA2* Y2H clones to test for interaction with TCO; however, the resulting proteins did not accumulate to an appreciable level (Figure 6B). Additionally, we tested the other nuclear-localized α subunit, AtCKA3, for interaction with TCO, and although this full-length subunit was efficiently expressed (Figure 6B), it failed to interact with TCO (Figure 6A). This indicates that TCO displays specificity regarding which CK2 subunits it interacts with.

Based on these Y2H results, we hypothesized that TCO may be phosphorylated by CK2. A minimum consensus sequence for CK2-mediated phosphorylation has been defined as SXXE/D, with S serving as the phospho-acceptor site [37]. The X residue adjacent to the phospho-acceptor S is also commonly an acidic residue [37]. Based on these characteristics, we identified a putative consensus CK2 target site in the middle of the TCO protein (S_75_ESD) (Figure 6C). To assess whether posttranslational modification of this serine is important for TCO function, we generated phospho-deficient (S75A) and phospho-mimic (S75D) TCO variants, fused these to GFP, and placed them under the control of the 2x35S CaMV promoter. Transient expression in tobacco leaves showed that both 2x35Sp::mTCO(S75A)-GFP and 2x35Sp::mTCO(S75D)-GFP products localized to the nucleus and variably accumulated in subnuclear foci (Figure 7). This closely resembled the distribution of 2x35Sp::TCO-GFP (Figure 2F,G and Figure 7), indicating that the mutation of S75 did not affect the expression or subcellular localization of TCO protein. We also generated stable 2x35Sp::mTCO(S75A)-GFP and 2x35Sp::mTCO(S75D)-GFP *Arabidopsis* transgenic lines, but found that these resembled plants overexpressing wild-type TCO (Figure 8).

Because protein sites targeted by CK2 can vary from the defined consensus sequence [37], it is possible that CK2 phosphorylates residues of TCO other than S75. To experimentally identify such positions, we purified glutathione S-transferase (GST)-tagged TCO from a bacterial expression system, incubated this with recombinant CK2 enzyme, and performed mass spectrometry to assess TCO phosphorylation. This analysis identified a number of TCO serine residues phosphorylated by CK2 that were not modified in mock-treated control samples, including S11, S12, S73 and S74 (Figure 6C and Figure 9).

To assess the biological relevance of these targeted serine residues, we designed various 2x35Sp::mTCO-GFP transgenes with phospho-deficient (S-to-A) or phospho-mimic (S-to-D) modifications. In all cases, these mTCO proteins accumulated to high levels and localized to the nucleus when transiently expressed in tobacco leaves (Figure 7), indicating that overall expression and subcellular distribution was not affected by the mutations. We also assessed whether stable transformation of these mTCO transgenes in *Arabidopsis* would result in phenotypes that differed from 2x35Sp::TCO-GFP. Stable *Arabidopsis* transformants were readily generated for the majority of the mTCO constructs, and these lines generally resembled 2x35Sp::TCO-GFP plants, as they lacked obvious seed patterning defects and displayed tightly curled leaves (Figure 8). Stable transgenic lines of 2x35Sp::mTCO(S11A,S12A,S73A,S74A)-GFP, however, could not be isolated despite repeated transformation attempts, and despite the fact that this mTCO transgene produced protein that accumulated to high levels when transiently expressed (Figure 7). This suggests that stable overexpression of this mTCO variant, which harbors multiple mutations that prevent CK2-mediated phosphorylation, compromises plant viability. At a mechanistic level, it remains to be determined how broadly preventing CK2-mediated phosphorylation of TCO negatively affects plant growth.

## 3. Discussion

The development of multicellular organisms relies on complex gene expression patterns established through transcriptional activation and repression. We have identified *TCO*, a putative transcriptional regulator in *Arabidopsis*, whose overexpression causes broad misregulation of reproductive genes in the vegetative stage, resulting in patterning abnormalities. This collection of defects, which includes curled leaves and early flowering, resembles mutants of chromatin-modifying factors [2,31,38], implicating a role for TCO in this mode of transcriptional regulation. This is further supported by the nuclear localization of TCO, which tends to accumulate in distinct foci or “speckles” reminiscent of known chromatin regulators [4,30,31,32,33]. Moreover, the enhancement and suppression of *tco-1D* developmental defects by *clf* and *ult1*, respectively, suggests that TCO may regulate a shared set of target genes with these well-characterized chromatin modifiers. While these genetic interactions were apparent in plants ectopically expressing *TCO* (Figure 4A-K; Appendix A), functional overlap may also occur in their normal domains of expression. Specifically, reporter gene analyses demonstrated that *TCO* acts during embryo/seed development, and both *CLF* and *ULT1* are expressed in developing seeds [34,39]. TCO may also interact with other chromatin modifiers to influence gene expression in these tissues. Indeed, a recent unpublished report describes a physical interaction between TCO (therein named EFC) and Multicopy Suppressor of Ira1 (MSI1) [40], a core component of *Arabidopsis* PRC2 complexes [5]. While these observations support a role for TCO in chromatin-dependent transcriptional regulation, the mechanism by which TCO participates in these processes is unclear, given its small size and lack of conserved functional domains.

Our determination that the protein kinase CK2 interacts with and phosphorylates TCO provides some insight into the nature of its activity. Phosphorylation of TCO could potentially affect its function in a variety of ways, including by altering its subcellular localization, its interaction with other proteins, and/or its association with chromatin [9]. Mutation of TCO serine residues targeted by CK2, however, did not alter its nuclear localization or speckle formation, suggesting that CK2-mediated phosphorylation influences other aspects of TCO activity. Similarly, overexpression of most TCO variants with either phospho-deficient or phospho-mimic mutations did not produce phenotypes that differed from plants overexpressing wild-type TCO. Here, functional differences between these various transgenes may have been obscured by their high expression levels. Support for this notion comes from analyses of the *Arabidopsis* bZIP transcription factor TGA2, which is also targeted by CK2. While mutation of TGA2 phosphorylated residues did not affect its nuclear localization, it compromised its ability to bind DNA [41]. Even so, overexpression of wild-type and mutated TGA2 activated target genes to the same extent in planta, leading to the suggestion that high protein levels masked functional disparity between the TGA2 variants [41]. That CK2-mediated phosphorylation of TCO is biologically relevant is suggested by our inability to isolate *Arabidopsis* transgenic lines overexpressing mTCO with all four targeted serine residues mutated to alanine (S11A,S12A,S73A,S74A), despite efficient transient expression of this same construct in tobacco cells. Although the reason for this is unclear, it is possible that phosphorylation of TCO modulates its transcriptional activity, and that overexpressing phospho-deficient TCO broadly disrupts gene expression to an extent that compromises plant viability.

TCO may interact with CK2 not only to mediate its own phosphorylation, but to recruit CK2 to protein complexes to target other factors. The use of protein–protein interaction as a means to modulate kinase activity has been previously proposed for CK2 [42,43], and the reported interaction between TCO and the PRC2 component MSI1 [40] could facilitate this type of mechanism. CK2-mediated phosphorylation of PRC factors has been reported to influence their protein–protein interactions, histone-binding properties and enzymatic activities [12,13,14,44], raising the possibility that mutation of *TCO* indirectly affects the ability of CK2 to target chromatin regulators. If the Enhancer of Zeste (E(z)) protein CLF is one such chromatin regulator, this could explain why *tco-1D* phenocopies *clf*. In a similar manner, phosphorylation of the human homolog of E(z), EZH2, interferes with its ability to bind and methylate histones, resulting in derepression of target genes [10].

Contrasting the broad gene misregulation in *tco-1D* seedlings, *tco* loss-of-function alleles affected gene expression to a lesser extent when tested in young silique tissue (Figure 5C,D). While this can be explained by the restricted domain of expression naturally displayed by *TCO*, the reason that both gain- and loss-of-function alleles displayed gene upregulation is unclear. Removing *TCO* function from its natural domains of expression could disrupt protein complexes that rely on TCO and/or CK2 activity, resulting in gene derepression. Strong, ectopic expression of *TCO* could similarly affect gene expression if this causes TCO to interfere with the formation of protein complex(es). Indeed, overexpression of factors that operate in protein complexes can mimic loss-of-function defects due to antimorphic effects [45]. Based on the reported binding of TCO to MSI1 [40], it is possible that high levels of TCO interfere with the ability of MSI1 to associate with other binding partners, including LIKE HETEROCHROMATIN PROTEIN 1/TERMINAL FLOWER 2 (LHP1/TFL2), which performs a Pc-like role in *Arabidopsis* [46,47]. Consistent with this scenario, *tco-1D* plants phenotypically resemble *lhp1*/*tfl2* loss-of-function mutants [31,48] and misexpress genes that rely on LHP1/TFL2-mediated repression such as *AG* and *SEP3* [49].

Notably, the most striking developmental defects caused by *TCO* overexpression arose in tissues that do not naturally express *TCO*, such as vegetative leaves. Growth responses that normally operate in the absence of TCO may, therefore, be particularly sensitive to artificial exposure to TCO activity. In these cases, TCO may engage in non-specific protein interactions, interfere with the composition of transcriptional regulatory complexes, and/or broadly impinge on chromatin topology, as suggested by widespread gene misregulation in *tco-1D* seedlings (Figure 3F; Appendix A). In contrast, *TCO* overexpression did not appear to affect tissues that normally express *TCO*, namely the seed (Figure 5A,B). Specifically, *tco-1D* and 2x35Sp::TCO-GFP transgenic backgrounds lacked obvious seed abnormalities, even though such defects were present in the loss-of-function allele *tco-1* (Appendix A). These observations indicate that growth processes that already involve TCO activity, such as seed development, may be able to accommodate further increases in TCO levels. Conversely, these same processes are likely to become compromised if *TCO* activity is reduced or abolished, as displayed by *tco-1* mutant seeds.

A future challenge will be to better clarify the specific endogenous role of *TCO* in *Arabidopsis* development. Notably, *TCO* has been identified as a candidate imprinted gene that displays seed-specific expression and hypomethylation in the endosperm [50]. Many imprinted genes play roles in endosperm/seed development [51], including a number of chromatin regulators [52]. It is plausible that *TCO* also contributes to this mode of transcriptional regulation during seed development, as our experimental data implicates *TCO* in the regulation of chromatin. *TCO*-like genes are also present in the genomes of other *Brassicaceae* species (Appendix A), and their functional characterization will provide further insight into the role of *TCO* and the extent to which this role is conserved in other plant species.

## 4. Materials and Methods

### 4.1. Plant Material

Plants were grown on soil in a growth chamber under a 16-hour light/8-hour dark cycle. The Landsberg *erecta* (L*er*) ecotype of *Arabidopsis thaliana* (L.) Heynh served as wild type, unless otherwise noted. Genetic analyses used the previously described mutants *ag-1* [53], *clf-2* [2], *ult1-1* and *ult1-2* [54]. PCR-based genotyping of *clf-2*, *ult1-1* and *ult1-2* alleles followed established protocols [4,55]. The *tco-1D* gain-of-function mutant was isolated from an activation-tagging screen of L*er* using T-DNA vector pSKI015 [56]. The *tco-1* and *tco-2* loss-of-function mutants are T-DNA insertion alleles (SALK_018803 and SALK_112041, respectively) [57]. Analyses involving these insertion mutants used the Columbia-0 (Col-0) ecotype of *Arabidopsis* as wild type.

### 4.2. Reverse Transcription-Polymerase Chain Reaction (RT-PCR)

An Invitrogen SuperScript first-strand synthesis system (Thermo Fisher Scientific, Waltham, MA, USA) was used for reverse transcription of total RNA samples purified using an RNeasy Plant Mini Kit (Qiagen Inc., Germantown, MD, USA). Quantitative real-time RT-PCR on cDNA samples was performed with a Mx3005P QPCR system (Agilent Technologies, Santa Clara, CA, USA) using PerfeCTa SYBR Green SuperMix (Quanta Biosciences Inc., Beverly, MA, USA). Data analysis was carried out with MxPro QPCR software (Agilent Technologies). Relative expression levels were normalized against *ACTIN7* [58]. Primer sequences used for cDNA amplification are provided in Appendix A.

### 4.3. Transgenic Plant Lines

The genomic location of the activation-tagging T-DNA in *tco-1D* was determined using TAIL PCR [15]. To generate 2x35Sp::TCO-GFP, the *TCO* gene was amplified by PCR and cloned into pCR-Blunt II-TOPO (Thermo Fisher Scientific). The *TCO* gene was then digested from this vector and introduced into the *Pst*I-*Hin*dIII sites of plasmid pBJ36 [59], placing *TCO* under the control of the 2x35S CaMV promoter and in-frame with the green fluorescent protein (GFP). This fusion gene was subcloned into the *Not*I site of binary vector pART27 [60] and transformed into the L*er* ecotype using the *Agrobacterium*-mediated floral dip method [61]. All 2x35Sp::TCO-GFP mutant variants were generated using the above strategy, modified as follows: mTCO(S75A) and mTCO(S75D) were produced by PCR-based site-directed mutagenesis, and mTCO(S11A,S73A), mTCO(S11D,S73D), mTCO(S11A,S12A,S73A,S74A) and mTCO(S11D,S12D,S73D,S74D) were ordered as custom-designed gene fragments (Integrated DNA Technologies, Inc., Skokie, IL, USA). 2x35Sp::YFP and all GFP reporter constructs were transiently expressed in tobacco leaves as previously described [62]. TCOp::TCO-GUS is a translational fusion between TCO and β-glucuronidase (GUS), flanked by the promoter (2.6 kb upstream of start codon) and 3′ non-coding regulatory sequence (1.5 kb downstream of stop codon) of the *TCO* gene. This fusion gene was assembled in pBJ36 and subcloned into pMLBART [59] for transformation into the *Arabidopsis* L*er* ecotype. All primer sequences used in PCR are provided in Appendix A.

### 4.4. Histology and Microscopy

Siliques of *Arabidopsis* plants expressing TCOp::TCO-GUS were GUS-stained for 24 h as previously detailed [58], fixed overnight in FAA (3.7% formaldehyde, 50% ethanol, 5% glacial acetic acid), embedded in paraffin, and sectioned to a thickness of 20 µm. Sections were then deparaffinized, rehydrated through a reverse ethanol series (100% to 30%), incubated in water, mounted and imaged. For embryo dissection, seeds were fixed in 4% paraformaldehyde under vacuum for 2 h, rinsed three times in phosphate buffered saline, and incubated in clearing solution (6M urea, 30% glycerol, 0.1% Triton X-100) [63] for 3 weeks at 4 °C. Embryos were liberated from the seed coat by applying gentle pressure. An Olympus SZX16 dissecting microscope was used to capture images of live plant tissues, while an Olympus BX61 compound microscope was used to capture images of fixed/sectioned tissues and GFP fluorescence (Olympus, Center Valley, PA, USA).

### 4.5. TCO Protein Expression and Analyses

Yeast two-hybrid screens using TCO as bait and an *Arabidopsis* floral cDNA library as prey were performed as previously described [62]. Directed yeast two-hybrid assays and Western blotting were conducted as previously reported [64]. GST protein fusions were expressed and purified as described [64]. Prior to incubation with recombinant CK2 enzyme (New England Biolabs, Inc., Ipswich, MA, USA), GST-TCO protein bound to glutathione agarose beads was washed twice with PBSI buffer (10 mM Na_2_HPO_4_, 2 mM KH_2_PO_4_, 2.7 mM KCl, 137 mM NaCl, 0.1% IGEPAL CA-620) and twice with accompanying Protein Kinase (PK) reaction buffer. 45 µL of PK buffer was added to the beads, followed by 1 µL of 10 mM ATP and 1000 U of CK2 enzyme. A mock GST-TCO reaction in the absence of CK2 enzyme was performed in parallel. The on-bead reactions were incubated at 30 °C for 1 h and subsequently washed twice in PBSI. GST-TCO protein was eluted with 20 mM reduced glutathione (50 mM Tris-HCl, pH 8), followed by a second elution in low salt buffer (150 mM NaCl, 50 mM Tris-HCl, pH 7.5, 1 mM EDTA) containing 0.05% *Rapi*Gest SF surfactant (Waters Corporation, Milford, MA, USA).

### 4.6. Mass Spectrometry Sample Preparation, Acquisition and Data Analysis

Eluted GST-TCO samples were prepared by diluting in 50 mM ammonium bicarbonate (pH 8) and digested with 1 μg/μL chymotrypsin at 37 °C overnight. Samples were then acidified by the addition of 10% trifluoroacetic acid to pH 2 and desalted in C18 stage tips. The peptides were resuspended in 4% formic acid and 3% acetonitrile, and approximately 2 μL of the 15 μL sample were loaded onto a 75 μm ID column packed with 25 cm of Reprosil C18 1.9 μm, 120 Å particles (Dr. Maisch High-Performance Liquid Chromatograph (HPLC) GmbH, Ammerbuch, Germany). Peptides were eluted into an Orbitrap Q-Exactive Plus mass spectrometer (Thermo Fisher) by gradient elution delivered by an Easy1200 nLC system (Thermo Fisher), using a gradient from 5% to 29% acetonitrile over 53 min. All mass spectrometry (MS) spectra were collected with orbitrap detection (70 K resolution). All data were searched simultaneously against the wild-type and mutant TCO protein sequences, as well as the yeast protein database (downloaded 13 January 2015) using MaxQuant [65]. Peptide, protein and phosphorylation sites identification was filtered to a 1% false-discovery rate. Spectra were annotated using Skyline [66]. The mass spectrometry data files (raw and search results) have been deposited to the ProteomeXchange Consortium (http://proteomecentral.proteomexchange.org) via the PRIDE partner repository [67] with the dataset identifier PXD011565.

## Figures and Tables

**Figure 1 ijms-20-00099-f001:**
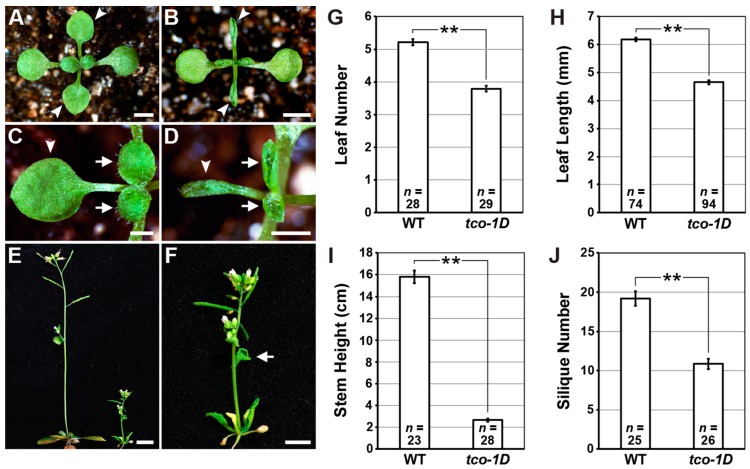
The activation-tagged mutant *tco-1D* has dominant, pleiotropic effects on *Arabidopsis* growth and development. (**A**–**D**) 14 days after germination (DAG) plants of wild type (**A**,**C**) and *tco-1D* (**B**,**D**). Arrowheads denote first pair of vegetative leaves, while arrows indicate the third and fourth leaves. Note strong upward curling displayed by *tco-1D* leaves. (**E**) 40 DAG plants of wild type (left) and *tco-1D* (right). (**F**) Higher magnification of *tco-1D* plant depicted in (**E**), with arrow denoting upward-curled cauline leaf. Bars: (**A**,**B**) 2 mm; (**C**,**D**) 1 mm; (**E**) 1 cm; (**F**) 5 mm. (**G**–**J**) Phenotypic comparisons between wild type (WT) and *tco-1D*. Number of vegetative leaves initiated prior to flowering (**G**), length of first vegetative leaves at 16 DAG (**H**), primary inflorescence stem height at 65 DAG (**I**), and number of siliques produced by the primary inflorescence stem (**J**) are shown. Data are represented as mean +/− standard error (SE). Statistically significant differences are indicated (** *p* < 0.001; two-tailed *t*-test). Sample size (*n*) of each genotype is provided.

**Figure 2 ijms-20-00099-f002:**
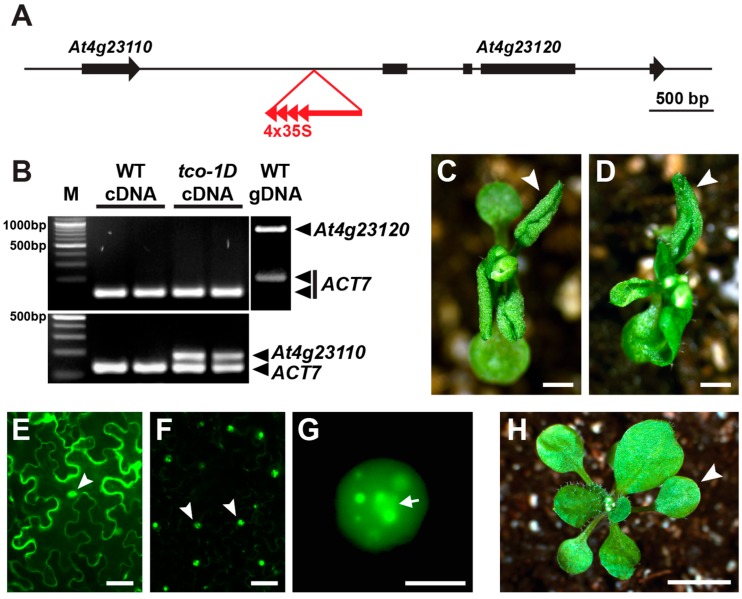
The *Arabidopsis* gene *At4g23110* is overexpressed by the activation-tagging T-DNA in *tco-1D*. (**A**) Schematic showing the insertion site of the activation-tagging T-DNA (red) in *tco-1D* between genes *At4g23110* and *At4g23120*. The 35S enhancer elements (red arrowheads) are oriented upstream. Gene exons are depicted as black rectangles, with black arrowheads showing gene orientation. (**B**) Semi-quantitative reverse transcription-polymerase chain reaction (RT-PCR) assessing the expression of genes *At4g23110* (bottom) and *At4g23120* (top) in 7 days after germination (DAG) seedlings of *tco-1D* and wild-type (WT), with *ACTIN7* (*ACT7*) serving as an internal control. Two biological replicates of each RT-PCR reaction are shown. A PCR with WT genomic DNA shows efficient amplification of *At4g23120* (top, right). (**C**) *tco-1D* and (**D**) 2x35Sp::TCO-GFP plants at 19 days after germination (DAG). Note similar curling of first vegetative leaves (arrowheads). (**E**–**G**) Transient transfections of tobacco leaves. (**E**) 2x35Sp::YFP is localized to the cell nucleus (arrowhead) and cytoplasm. (**F**) 2x35Sp::TCO-GFP localizes to nuclei (arrowheads). (**G**) Close-up of 2x35Sp::TCO-GFP nucleus showing localization in a central subnuclear compartment predicted to be nucleolus (arrow) and in other punctate nuclear foci. (**H**) WT plant at 19 DAG with first vegetative leaf denoted (arrowhead). Bars: (**C**,**D**) 1 mm; (**E**,**F**) 50 μm; (**G**) 10 μm; (**H**) 5 mm.

**Figure 3 ijms-20-00099-f003:**
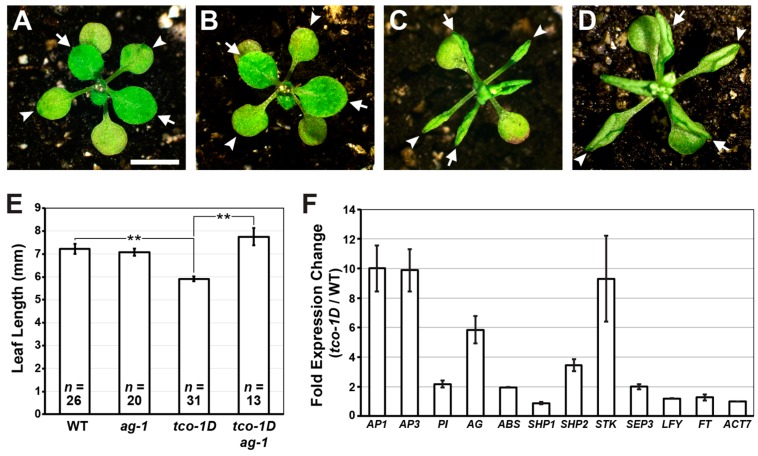
Developmental defects of *tco-1D* are associated with misregulation of multiple floral organ identity genes in *Arabidopsis*. (**A**–**D**) 20 days after germination (DAG) plants of wild type (**A**), *ag-1* (**B**), *tco-1D* (**C**) and *tco-1D ag-1* (**D**). Arrowheads denote first pair of vegetative leaves, while arrows indicate third and fourth leaves. Bars: 5 mm. (**E**) Length of third/fourth vegetative leaves of wild type (WT), *ag-1*, *tco-1D* and *tco-1D ag-1* plants. Data are represented as mean +/− SE. Asterisks denote statistically significant differences (** *p* < 0.001; two-tailed t-test). Sample size (*n*) of each genotype is given. (**F**) Quantitative RT-PCR on genes controlling floral organ identity, floral meristem identity and flowering time in *tco-1D* seedlings (7 DAG) relative to wild type (WT). Assessed genes are *APETALA1* (*AP1*), *APETALA3* (*AP3*), *PISTILLATA* (*PI*), *AGAMOUS* (*AG*), *ARABIDOPSIS B SISTER* (*ABS*), *SHATTERPROOF1* (*SHP1*), *SHATTERPROOF2* (*SHP2*), *SEEDSTICK* (*STK*), *SEPALLATA3* (*SEP3*), *LEAFY* (*LFY*), and *FLOWERING LOCUS T* (*FT*). Fold expression change (*tco-1D*/WT) was normalized against *ACTIN7* (*ACT7*) expression. Data are represented as mean +/− SE of two biological replicates. Primer sequences are provided in Appendix A.

**Figure 4 ijms-20-00099-f004:**
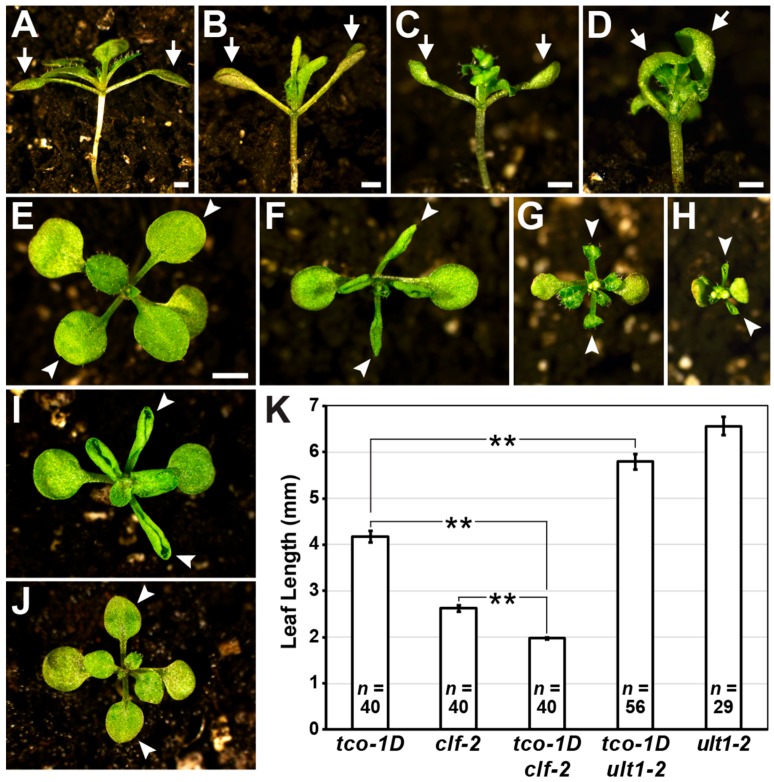
Vegetative defects of *tco-1D* are enhanced by *clf* and suppressed by *ult1*. (**A**–**D**) Lateral views of 17 days after germination (DAG) wild type (WT) (**A**), *tco-1D* (**B**), *clf-2* (**C**) and *tco-1D clf-2* (**D**) plants, with cotyledons denoted (arrows). (**E**–**J**) Apical views of 17 DAG WT (**E**), *tco-1D* (**F**), *clf-2* (**G**), *tco-1D clf-2* (**H**), *tco-1D ult1-2* (**I**) and *ult1-2* (**J**) plants, with first vegetative leaves denoted (arrowheads). Bars: (**A**–**D**) 1 mm; (**E**–**J**) 2 mm. (**K**) Lengths of first vegetative leaves of *tco-1D*, *clf-2*, *tco-1D clf-2*, *tco-1D ult1-2* and *ult1-2* are shown. Data are represented as mean +/− SE. Asterisks denote statistically significant differences (** *p* < 0.001; two-tailed *t*-test). Sample size (*n*) of each genotype is provided.

**Figure 5 ijms-20-00099-f005:**
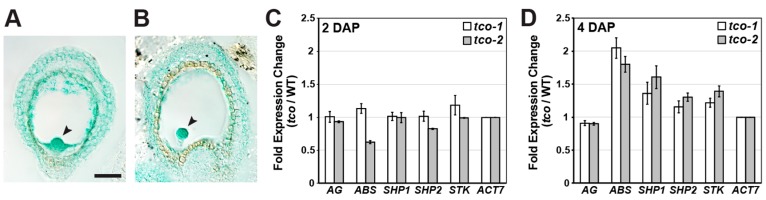
*TCO* is expressed throughout the *Arabidopsis* seed where it affects MADS-box gene expression. (**A**,**B**) Expression of TCOp::TCO-GUS in seeds of two independent transgenic lines of *Arabidopsis*. Note expression throughout the seed coat, endosperm and globular stage embryos (arrowheads). Bars: 50 μm. (**C**,**D**) Quantitative RT-PCR on C-class (*AG*) and D-class genes (*ABS*, *SHP1*, *SHP2*, *STK*) in *tco-1* and *tco-2* siliques relative to wild type (WT) at 2 days after pollination (DAP) (**C**) and 4 DAP (**D**). Fold expression change (*tco*/WT) was normalized against *ACTIN7* (*ACT7*) expression. Data are represented as mean +/− SE of two biological replicates.

**Figure 6 ijms-20-00099-f006:**
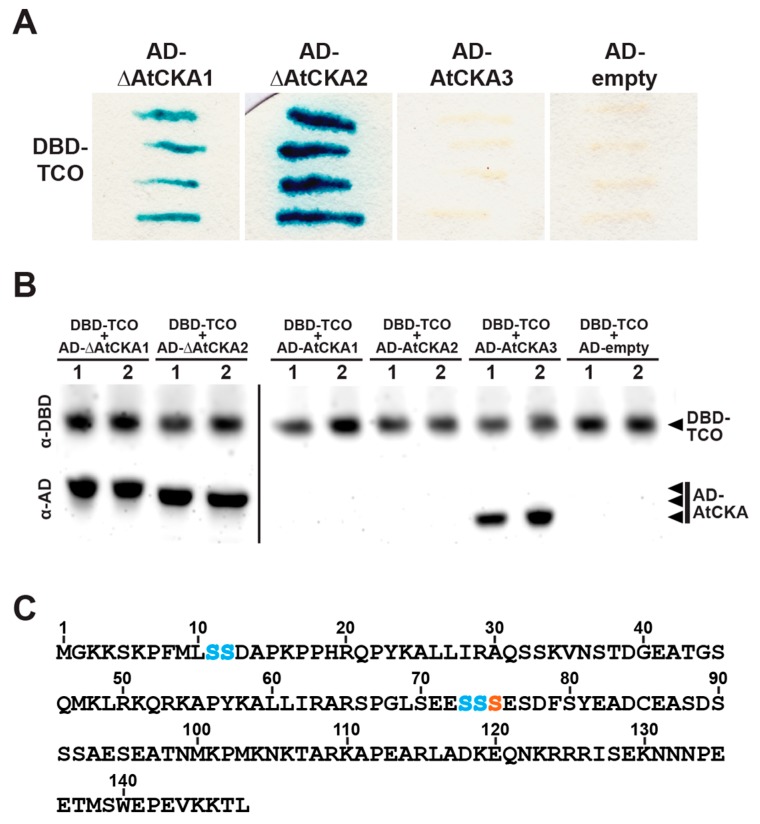
TCO protein interacts with catalytic subunits of the protein kinase CK2. (**A**) Yeast two-hybrid assays testing interaction between TCO (fused to the GAL4 DNA-binding domain (DBD)) and ∆AtCKA1(aa25-409), ∆AtCKA2(aa32-403) and AtCKA3(full-length) (each fused to the GAL4 activation domain (AD)), and an empty AD vector control are depicted. Positive interactions are indicated by blue coloration. (**B**) Western blot showing expression levels of yeast two-hybrid constructs. DBD-TCO, AD-∆AtCKA1(aa25-409), AD-∆AtCKA2(aa32-403) and AD-AtCKA3(full-length) show efficient expression in yeast. Full-length AD-AtCKA1 and AD-AtCKA2 protein does not accumulate to appreciable levels. The AD-empty vector control protein is not visible due to its small size. Two independent yeast transformants are shown for each vector combination. (**C**) Amino acid sequence of TCO. Serine (S) residues that reside within a putative consensus CK2 recognition sequence (SXXE/D) (orange) or that were experimentally determined to be modified by CK2 in this study (blue) are indicated.

**Figure 7 ijms-20-00099-f007:**
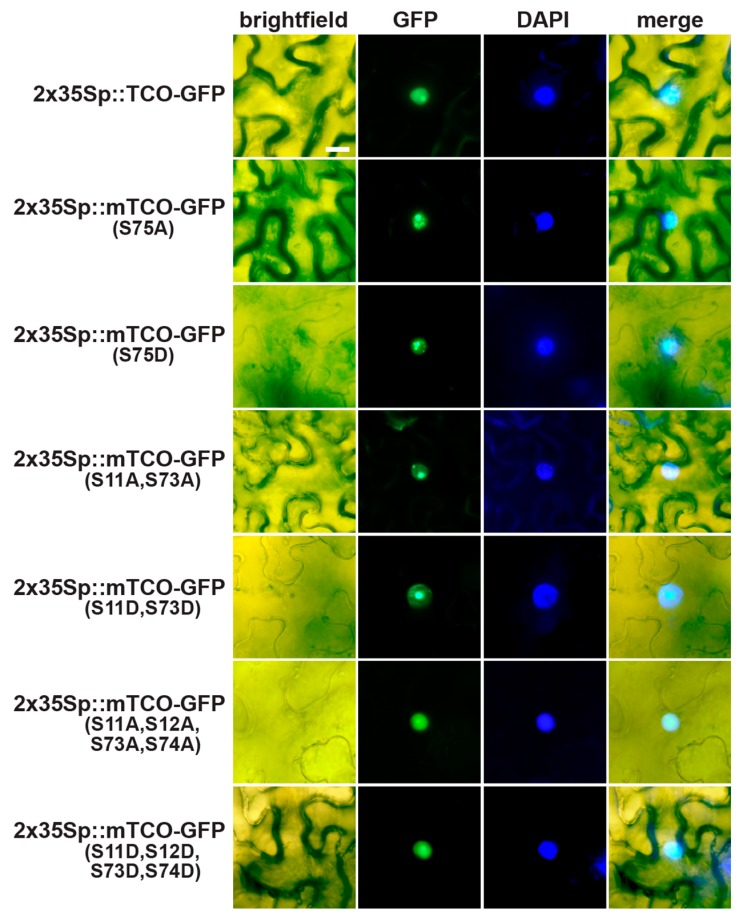
Phospho-deficient (S-to-A) and phospho-mimic (S-to-D) mutations of TCO do not alter protein subcellular localization. Overexpression constructs of wild-type TCO (2x35Sp::TCO-GFP) and mutated TCO variants (2x35Sp::mTCO-GFP) were transiently expressed in tobacco leaves. Proteins from all constructs accumulated to comparable levels (based on GFP fluorescence), localized to the nucleus, and variably displayed subnuclear speckles. Brightfield, GFP, and DAPI (nuclear stain) images are shown, along with a merged overlay of all three. Bars: 20 μm.

**Figure 8 ijms-20-00099-f008:**
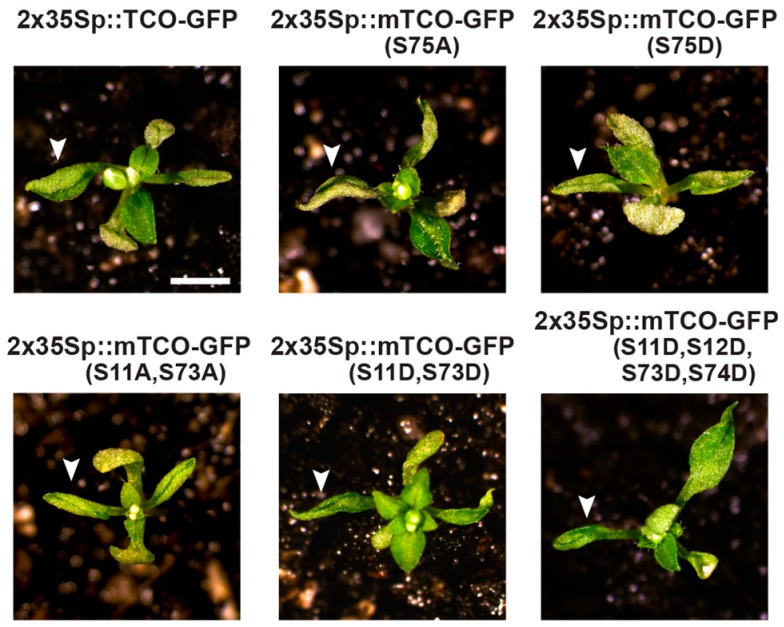
Phenotypic effects of overexpressing TCO-GFP transgenes in *Arabidopsis*. At 23 days after germination, primary transformants of phospho-deficient and phospho-mimic 2x35Sp::mTCO-GFP transgenes resembled those of 2x35Sp::TCO-GFP, all of which displayed short, curled leaves (arrowheads) and flowered early. The lone exception was 2x35Sp::mTCO(S11A,S12A,S73A,S74A)-GFP for which primary transformants could not be isolated.

**Figure 9 ijms-20-00099-f009:**
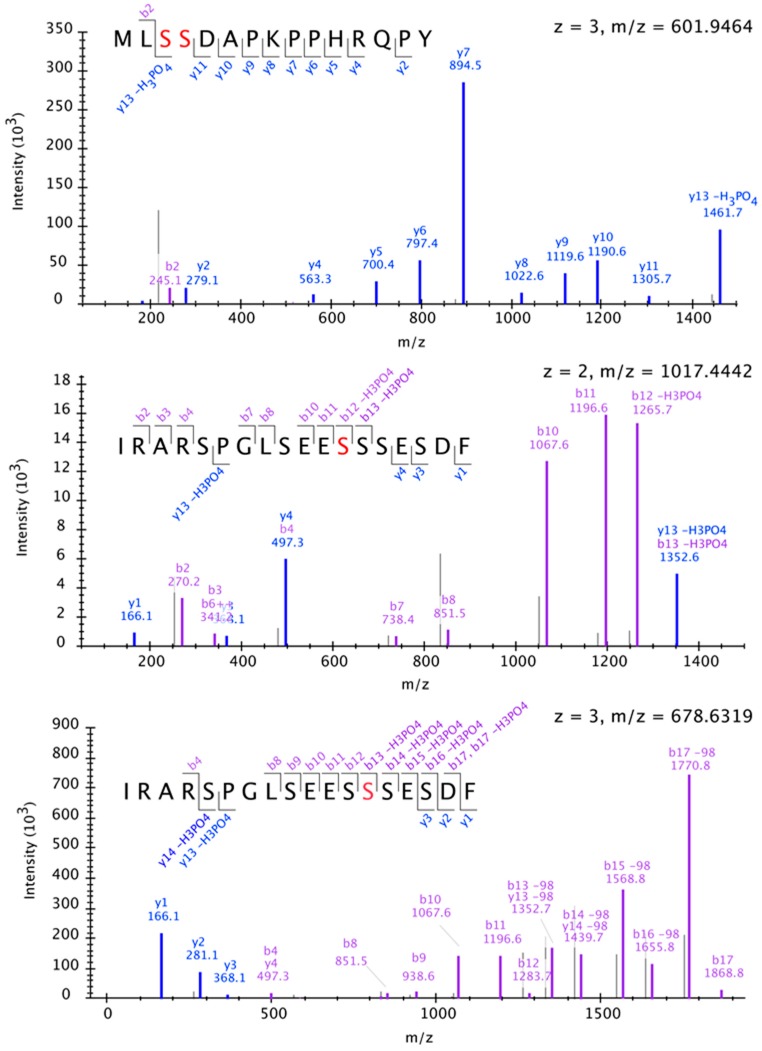
Mass spectrometry analysis of TCO residues phosphorylated by CK2. Representative annotated spectra for the phosphorylation sites identified are shown. Observed fragment ions are annotated in the displayed peptide sequences, with phosphorylation sites indicated by red-colored amino acids. Each peptide contains only a single phosphorylation site; however, in the case of the peptide MLSSDAPKPPHRQPY (top), the exact site of phosphorylation is ambiguous between two adjacent serine residues.

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
