# Peer review of "TCO, a Putative Transcriptional Regulator in Arabidopsis, Is a Target of the Protein Kinase CK2"

_ijms, 2018, doi:10.3390/ijms20010099_

Round 1
Reviewer 1 Report
The manuscript by Weinman et al. is focused on TCO, a putative transcriptional regulator in Arabidopsis, which is identified by authors as a target of the protein kinase CK2. The manuscript contains a set of novel valuable data. However, I have some concerns:
- Authors conclude that TCO-GFP is localized in nucleoli. I agree that the structure pointed by an arrow on Fig 3G most likely represents a nucleolus. Nonetheless, this is still a hypothesis, which should have an experimental support. For example, co-localization experiments with fluorescent protein fused to protein localized to nucleolus can support the conclusion.
- In my opinion subsections within the ‘Results’ section should correspond to their ‘own’ figure panels. Thus, I would like to recommend authors to recombine their figure panels presented in the current version in accordance with a sequence of subsections.
- Abbreviations of the genes within Fig. 2F should be explained within the figure legend.
- The references to some figures are absent in the text.
Author Response
Please find below our responses (in bold type) to each Reviewer comment. We have made every effort to address all suggestions as best as possible. We thank the Reviewer for their valuable comments and acknowledge that the incorporation of their suggestions has improved our manuscript.
Response to Reviewer 1 Comments:
Point 1 - Authors conclude that TCO-GFP is localized in nucleoli. I agree that the structure pointed by an arrow on Fig 3G most likely represents a nucleolus. Nonetheless, this is still a hypothesis, which should have an experimental support. For example, co-localization experiments with fluorescent protein fused to protein localized to nucleolus can support the conclusion.
Response 1 – We agree with this comment and have changed or removed all phrases that had implied that the subnuclear compartment showing TCO accumulation was definitively the nucleolus. On the revised version of our manuscript, this includes:
Line 139 – changed to “subnuclear compartment predicted to be nucleolus (arrow)”
Line 429 – removed phrase “and shows enrichment in nucleolus”
Line 463 – removed phrase “were enriched in the nucleolus”
Line 474 – removed phrase “with nucleolar enrichment”
Line 497 – removed phrase “with nucleolar enrichment”
We agree that co-localization experiments could definitively identify the subnuclear compartment as the nucleolus. However, suitable transgenic lines cannot be established within the requested time frame for manuscript re-submission. Therefore, we have changed the text as described above (as also suggested by Reviewer 3).
Point 2 - In my opinion subsections within the ‘Results’ section should correspond to their ‘own’ figure panels. Thus, I would like to recommend authors to recombine their figure panels presented in the current version in accordance with a sequence of subsections.
Response 2 – In accordance with this request, we have reorganized the Results section such that all figure panels are self-contained within individual subsections. For example, parts of Figure 3 (which was Figure 2 in the original submission) no longer span multiple Results subsections, and instead are now entirely contained within subsection 2.2.
Point 3 - Abbreviations of the genes within Fig. 2F should be explained within the figure legend.
Response 3 – As requested, we have defined all gene abbreviations in the figure legend of Figure 3F (which was Figure 2F in the original submission).
Point 4 - The references to some figures are absent in the text.
Response 4 – We have carefully reviewed the entire manuscript to ensure that all panels of every figure are referenced at least once in the text.
Reviewer 2 Report
The work from Weinman et al., describes the identification of a putative transcriptional regulator, here named TACO leaf that regulates different stages of plant development.
The work is interesting but in order to be scientifically sound, few more experiments are required to corroborate the conclusions.
Fig1G: Is it possible that plants have only 5 to 6 leaves before flowering? Are you considering only rosette leaves or cauline leaves as well?
Fig.2F I would prefer that the expression data will show also the wild type expression of these genes rather than being normalised to their wild type levels. Besides what is the control gene that has been measured to define the targets expression?
Fig.3B RT-PCR is not enough. Authors need to do qRT-PCR to prove that TACO leaf gene is up-regulated in the mutant. Besides the expression of the other targets might be also be detectable by qRT-PCR.
Fig3F to G Tobacco images require white field. What about the GFP localisation Arabidopsis stable lines?
Fig5 Suddenly the authors describe knock-out mutants for the TACO gene and their effect in seed development. What about the phenotype of the over expressing lines in seed development and of the knock-out mutants for flower development? It is rather confusing that different lines are described for different developmental stages.
Fig7 I would suggest the authors to use better resolution images if they want to demonstrate that the mutations do not affect nuclear localisation.
Fig.8 Again, what about the effect of these lines on seed development? Since no major effect is observed compared to the wild type over-expressing lines, it is important to describe other phenotypes.
Minor point: a word is missing in line 36.
Author Response
Please find below our responses (in bold type) to each Reviewer comment. We have made every effort to address all suggestions as best as possible. We thank the Reviewer for their valuable comments and acknowledge that the incorporation of their suggestions has improved our manuscript.
Point 1 - Fig1G: Is it possible that plants have only 5 to 6 leaves before flowering? Are you considering only rosette leaves or cauline leaves as well?
Response 1 – Under our growth conditions (Long Day light regime), wild type Ler plants routinely flower after the production of 5 to 6 vegetative (rosette) leaves. These leaf numbers are in accordance with other published work describing the growth of Ler plants under similar conditions (for example, please refer to Reeves and Coupland, Plant Physiology, 2001, 126(3):1085-1091).
Nonetheless, to address this question, we have changed Line 92 of our modified manuscript from “the number of vegetative leaves initiated” to “the number of rosette leaves initiated” to emphasize that we are quantifying rosette leaves, and that cauline leaves are not included or considered in our leaf counts.
Point 2 - Fig.2F I would prefer that the expression data will show also the wild type expression of these genes rather than being normalised to their wild type levels. Besides what is the control gene that has been measured to define the targets expression?
Response 2 – In accordance with this request, we have added a supplementary figure (Figure S4) which complements Figure 3F (originally Figure 2F). Specifically, Figure S4 displays the gene expression data of both wild-type and tco-1D backgrounds individually (rather than normalizing tco-1D against wild-type levels).
Secondly, in the legends of both Figure 3F and Figure S4, we have made sure to indicate that the control gene used to normalize/measure the expression of the other test genes is ACTIN7 (ACT7).
Point 3 - Fig.3B RT-PCR is not enough. Authors need to do qRT-PCR to prove that TACO leaf gene is up-regulated in the mutant. Besides the expression of the other targets might be also be detectable by qRT-PCR.
Response 3 – As requested, we have added a supplementary figure (Figure S1) in which we perform qRT-PCR on genes At4g23110 and At4g23120 in tco-1D and wild-type backgrounds. The results from this analysis are in general agreement with data presented in Figure 2B (originally Figure 3B). Specifically, the qRT-PCR shows that At4g23110 is the only gene that is strongly and reliably upregulated in tco-1D vegetative tissues. The description and interpretation of this new data can now be found on lines 121-123 of the revised manuscript.
Point 4 - Fig3F to G Tobacco images require white field. What about the GFP localisation Arabidopsis stable lines?
Response 4 – Brightfield images were not captured for samples depicted in Figure 2F,G (originally Figure 3F,G). However, the same transgenic line (2x35Sp::TCO-GFP) is shown as a control on the top of Figure 8. In this case, brightfield images, DAPI stains (to stain nuclei) and image overlays are provided, clearly showing nuclear localization of this transgenic line.
Secondly, to address this request, we have added a new supplementary figure (Figure S3) which shows 2x35Sp::TCO-GFP distribution in stable Arabidopsis transgenic lines. The images show punctate subcellular accumulation consistent with nuclear localization, complementing the data from transient tobacco expression. Description of this new data can now be found on lines 160-162 of the revised manuscript.
Point 5 - Fig5 Suddenly the authors describe knock-out mutants for the TACO gene and their effect in seed development. What about the phenotype of the over expressing lines in seed development and of the knock-out mutants for flower development? It is rather confusing that different lines are described for different developmental stages.
Response 5 – Despite the fact that the tco-1 insertion allele displays seed patterning abnormalities, neither tco-1D nor 2x35Sp::TCO-GFP lines exhibit similar abnormalities. To clarify and emphasize that there are no seed defects associated with TCO overexpression, we have added information to the revised manuscript. In close proximity to Figure 5, this information can be found on lines 405-408 (and reinforced later on lines 502-504).
Point 6 - Fig7 I would suggest the authors to use better resolution images if they want to demonstrate that the mutations do not affect nuclear localisation.
Response 6 – After re-examining the PDF proof of our manuscript that was automatically generated for peer review, we agree with Reviewer #2 that the resolution of Figure 7 was compromised. However, the original images (which we expect will be used for publication) have a resolution of over 570 ppi (pixels per inch) and are far more clear. Therefore, to address this comment, higher-resolution images will indeed be used in the actual published version that will more clearly depict nuclear localization.
Point 7 - Fig.8 Again, what about the effect of these lines on seed development? Since no major effect is observed compared to the wild type over-expressing lines, it is important to describe other phenotypes.
Response 7 – Similar to our Response 5 above, the 2x35Sp::TCO-GFP lines do not exhibit seed patterning abnormalities. Therefore, to reinforce the new information presented on lines 405-408, we have also added clarifying information near Figure 8 (on lines 502-504) to emphasize that there are no seed defects associated with TCO overexpression.
Point 8 - Minor point: a word is missing in line 36.
Response 8 – We have added the word “that” to line 36, which now reads: “On a broad scale, this balance ensures that developmental programs are invoked at appropriate stages of an organism’s life cycle.”
Reviewer 3 Report
In this work Weinman et al. show functional characterisation of putative transcriptional regulator in Arabidopsis TACO LEAF (TCO). Firstly, the authors identify and describe an activation - tagged allele tco-1D, displaying upward-curled leaves and ectopic expression of floral identity genes including AG and others. ag mutation supressed tco-1D phenotype, suggesting that AG misexpression contributes to the observed phenotype in tco-1D. Next, they identify At4g23110 as TCO gene based on gene expression analysis and similar phenotype of At4g23110-GFP overexpressing line. The authors notice that characteristic phenotypes of tco-1D that resemble those of chromatin regulatory mutants including PcG, and test genetic interactions with clf (PcG) and ult (TrxG) mutants. TCO was found to be expressed in seeds, and seed development is affected in one of the analysed TCO insertional mutants. In the second part of the manuscript authors show that TCO protein interacts with CK2 kinase and map 4 phosphorylated serines located within two motifs after in vitro. Finally, they show that overexpression of TCO harboring phospho-deficient or phospho-mimic mutations does not lead to different phenotypes than overexpression of WT TCO, except that line expressing TCO with all four phospho-deficient mutations could not be isolated, suggesting it compromises plant viability. Together, although interpretation of some results was difficult, the work is of high quality and provides interesting and novel information about so far uncharacterised transcriptional regulator. I have a few comments that can be found below:
1) phenotypes observerved in two insertional mutants were different and the strong effects not visible in tco-2. It may suggest that this mutant is not a full knock-out. Can Authors provide qRT-PCR results od detecting transcripts upstream and downstream insertion positions ?
2) the conclusion about nucleolar localisation of TCO should be tuned-down, as Authors did not use any nucleolus marker in their analyses
3) Do tco-1D and TCO-GFP plants have any defects in seed development ? This information should be provided at least in discussion.
4) information about expression profile of CK2 should be provided.
5) Data on the fig. 9 need better description, for example peaks corresponding to phosphorylated residues should be marked as well as mass differences between the peaks.
In addition, I would consider changes in the order of the presented results so that the identification of the TCO gene is before further functional characterisation.
Author Response
Please find below our responses (in bold type) to each Reviewer comment. We have made every effort to address all suggestions as best as possible. We thank the Reviewer for their valuable comments and acknowledge that the incorporation of their suggestions has improved our manuscript.
Point 1 - phenotypes observerved in two insertional mutants were different and the strong effects not visible in tco-2. It may suggest that this mutant is not a full knock-out. Can Authors provide qRT-PCR results od detecting transcripts upstream and downstream insertion positions ?
Response 1 – We have compiled a new supplementary figure (Figure S6C) that directly addresses this request. This figure presents qRT-PCR data assessing TCO expression levels using primer pairs that bind either upstream or downstream of the T-DNA insertion sites of tco-1 and tco-2. The data suggest that low-levels of TCO transcripts are present in both tco insertion mutants, with tco-1 showing greater reductions in TCO levels compared to tco-2. A description about this new data can be found on lines 374-385 of our revised manuscript. Furthermore, as a result of these enlightening experiments, we have modified our interpretation as to why tco-1 and not tco-2 exhibits seed patterning defects. This interpretation can be found on lines 403-408 of the revised manuscript.
Point 2 - the conclusion about nucleolar localisation of TCO should be tuned-down, as Authors did not use any nucleolus marker in their analyses
Response 2 – We agree with Reviewer 3, and with Reviewer 1 who made the same comment. In accordance with these suggestions, we have changed or removed all phrases that had implied that the subnuclear compartment showing TCO accumulation was definitively the nucleolus. On the revised version of our manuscript, this includes:
Line 139 – changed to “subnuclear compartment predicted to be nucleolus (arrow)”
Line 429 – removed phrase “and shows enrichment in nucleolus”
Line 463 – removed phrase “were enriched in the nucleolus”
Line 474 – removed phrase “with nucleolar enrichment”
Line 497 – removed phrase “with nucleolar enrichment”
Point 3 - Do tco-1D and TCO-GFP plants have any defects in seed development ? This information should be provided at least in discussion.
Response 3 - Despite the fact that the tco-1 insertion allele displays seed patterning abnormalities, neither tco-1D nor 2x35Sp::TCO-GFP lines exhibit similar abnormalities. To clarify and emphasize that there are no seed defects associated with TCO overexpression, we have added information to the revised manuscript. This information can be found on lines 405-408, and reinforced later on lines 502-504.
Point 4 - information about expression profile of CK2 should be provided.
Response 4 – It has been experimentally determined that all CK2 subunits are broadly expressed throughout plant development, including in roots, vegetative leaves, reproductive stems and flowers. We have added this information (along with the appropriate citation) on lines 416-418 of the revised manuscript.
Point 5 - Data on the fig. 9 need better description, for example peaks corresponding to phosphorylated residues should be marked as well as mass differences between the peaks.
Response 5 – To address this request, we have modified Figure 9 significantly. Specifically, on each analyzed peptide, we have annotated observed fragment ions, and highlighted residues that are targeted by phosphorylation. We feel that these added descriptions have increased the clarity and meaning of our mass spectrometry data.
Point 6 - In addition, I would consider changes in the order of the presented results so that the identification of the TCO gene is before further functional characterisation.
Response 6 – As requested, we have changed the order of our figures so that the identity of the TCO gene is provided prior to detailed functional characterization. Specifically, our figure describing the identification of At4g23110 as the affected gene in tco-1D (previously Figure 3) is now presented before our figure characterizing floral gene misregulation in tco-1D (previously Figure 2).
Round 2
Reviewer 2 Report
The authors in this revised version addressed most of the concerns from the previous manuscript. However few points need to be addressed in a deeper manner:
Point 3: Showing qRT-PCR is good but I still think that the data should be shown without normalisation therefore the expression of both genes in the wild type and the taco 1-D mutant. Sup fig. 1B is not so informative, besides the the presence of double peaks for one of the PCR products could also indicate that the chosen primers are not so specific.
Point 5 and 7: If the over expressing lines don't show phenotype in seed germination, that needs to be discussed in more details. Also the word lines reported in the answers do not correspond at the content in the manuscript, most likely to do with the corrections.
Author Response
Point 3 - Showing qRT-PCR is good but I still think that the data should be shown without normalisation therefore the expression of both genes in the wild type and the taco 1-D mutant. Sup fig. 1B is not so informative, besides the the presence of double peaks for one of the PCR products could also indicate that the chosen primers are not so specific.
Response - To accommodate this request, we have added a qRT-PCR graph to our first Supplementary Figure (Figure S1A) which displays At4g23110 and At4g23120 gene expression data in wild-type and tco-1D backgrounds individually (rather than normalizing tco-1D against wild-type levels). This representation of the data further supports that At4g23110 is the upregulated gene in tco-1D.
Additionally, on the right side of Figure 2B, a PCR reaction using genomic DNA as template and the same At4g23120 primers employed in Figure S1 shows the amplification of a strong, specific band. Therefore, we believe that the weak, double peaks displayed by the At4g23120 dissociation curve (Figure S1C) is reflective of low/undetectable expression and not indicative of non-specific primers.
Point 5 and 7 - If the over expressing lines don't show phenotype in seed germination, that needs to be discussed in more details. Also the word lines reported in the answers do not correspond at the content in the manuscript, most likely to do with the corrections.
Repsonse – To address this request, we have added an entire paragraph to the Discussion section (second last paragraph) dedicated to examining plausible reasons why Arabidopsis lines that overexpress TCO do not show obvious seed defects, even though the loss-of-function insertion line tco-1 displays seed abnormalities.
In addition, we apologize for the word lines in our previous responses not aligning properly with the manuscript content. We suspect that the conversion of our WORD document to an Adobe PDF modified the numbering system.